

# Association between NKILA and some apoptotic gene expression in atherosclerosis

Burcu Bayyurt[1], Şeyda Akın[1], Nil Özbilüm Şahin[2] and İzzet Yelkuvan[1]

[1] Department of Medical Biology, Faculty of Medicine, Sivas Cumhuriyet University, Sivas, Turkey
[2] Department of Molecular Biology and Genetics, Faculty of Science, Sivas Cumhuriyet University, Sivas, Turkey

## ABSTRACT

Oxidized light-density lipoprotein (ox-LDL) causes endothelial dysfunction, which is an important determinant of atherogenesis, and subsequently leads to apoptosis. Atherosclerosis is one of the most significant cardiovascular diseases (CVDs) threatening human health and causes death worldwide. Recently, long noncoding RNAs (lncRNAs) have been suggested to involved in vascular biology. Ox-LDL activates nuclear factor kappa-B (NF-κB), and NF-κB interacting lncRNA (NKILA) inhibits NF-κB signaling. In this study, the hypothesis is that NKILA may regulate endothelial cell (EC) apoptosis and, therefore, play a role in the pathogenesis of atherosclerosis. This hypothesis is based on the knowledge that EC apoptosis contributes to atherosclerosis development and that NKILA has become a prominent lncRNA in CVDs. The expression of Bcl-2-associated X protein (BAX), caspase 9 (CASP9), cytochrome c (Cyt c, CYCS), apoptotic protease activating factor 1 (APAF1), and B-cell lymphoma 2 (BCL-2) genes in human umbilical vein endothelial cells (HUVEC) treated with ox-LDL and transfected with NKILA siRNA was analyzed using quantitative reverse transcription polymerase chain reaction (RT-qPCR). BAX, CASP9, CYCS, APAF1, and BCL-2 gene expression was downregulated in ox-LDL and NKILA siRNA-treated HUVEC. In addition, when threshold/quantification cycle (Cq) values of NKILA gene expression increased, Cq values of BAX, CASP9, APAF1, and BCL-2 gene expression increased statistics significantly. The expression detection of all these genes, resulting from NKILA gene silencing, may provide guidance for epigenetic studies on EC apoptosis in atherosclerosis.

# INTRODUCTION

Atherosclerosis, a chronic multifactorial disease, is closely associated with the development of cardiovascular diseases (CVDs) and is an important universal reasonfor death. Research so far has revealed that EC apoptosis constitutes the first step in atherosclerosis. Preventing excessive endothelial cell (EC) apoptosis may be a goal for atherosclerosis therapy (*Duan et al., 2021*). Oxidized light-density lipoprotein (ox-LDL) is a biomarker and has a critical role in the diagnosis and development of atherosclerosis (*Maggi et al., 1993*). Ox-LDL induces endothelial cell (EC) apoptosis, alters the secretory activity of ECs, and causes endothelial dysfunction (*Mitra et al., 2011*). Apoptosis is a programmed cell death that

Corresponding author
Burcu Bayyurt,
ebayyurt@yahoo.com.tr

appears in multicellular organisms to sustain homeostasis (*Kerr, Wyllie & Currie, 1972*). It is driven by two distinct pathways: intrinsic (mitochondrial) and extrinsic pathways that assemble to activate cysteine-aspartic proteases (caspases), which cause the degradation of proteins, and resulting the death of cells (*Wajant, 2002*). The mitochondrial pathway results from different stimuli in the absence of receptor–ligand interactions and is mediated by incremented mitochondrial membrane permeability. This leads to a change in the balance between anti-apoptotic and pro-apoptotic proteins of the B-cell lymphoma 2 (BCL-2) family. Mitochondrial permeability causes the release of contents of the mitochondria, including cytochrome c (Cyt c, CYCS), resulting in the activation of caspase and death of the cell (*Tait & Green, 2010*). After permeabilization of the mitochondria, Cyt c binds to Apaf-1. This induces conformational transformation and oligomerization of Apaf-1 to promote apoptosome formation. Apoptosome is composed of Apaf-1, CASP9, and Cyt c (*Tait & Green, 2010*). Cyt c activates the Apaf-1 as well as pro-caspase-9, which can then result in caspase 9 (CASP9) activation (*Hill et al., 2004*). CASP9 is a key trigger caspase involved in intrinsic apoptosis  (*Lee et al., 2017*). As a result, DNA fragmentation and chromatin condensation are induced during the late phase of apoptosis in a caspase-independent manner (*Joza et al., 2001*). Bcl-2-associated X protein (BAX) is a pro-apoptotic protein and is involved in mitochondrial damage and Cyt c release mechanisms (*O'Brien & Kirby, 2008*). Accumulating research highlights that lncRNAs have a pivotal role in vascular disorders, including atherosclerosis (*Vacante et al., 2021*; *Fasolo et al., 2021*; *Dong et al., 2021*; *Hu et al., 2019*). The NF-κB interacting lncRNA (NKILA) binds the NF-κB/IκB complex to hide phosphorylation sites of IκB, thereby stabilizing the NF-κB/IκB complex. Our current study is the first to analyze the expression of the pro-apoptotic genes such as BAX, CASP9, CYCS, and APAF1 and anti-apoptotic genes such as BCL-2 together in HUVEC treated with ox-LDL and NKILA siRNA. In this context, the expression level of genes involved in the mitochondrial pathway was examined in HUVEC lines treatment with ox-LDL, which is known as an apoptosis-inducing agent in ECs, and then transfected with NKILA siRNA. We investigated how silencing of the lncRNA NKILA, a known suppressor of nuclear factor kappa-B (NF-κB) activation, alters the expression of these genes involved in EC apoptosis.

## MATERIALS AND METHODS

### Cell line

Human umbilical vein endothelial cells (HUVECs) (ATCC) were received, thawed and passaged. Firstly, cells were cultured with human endothelial cell growth medium (DMEM supplemented with 10% FBS and 1% penicillin-streptomycin), and passages 4 and 5 were used for experiments. All cells were incubated at 37 °C in 5% $CO_2$ under 95% relative humidity. The cells were removed from the culture medium with 0.25% trypsin every 2–3 days and subcultured, and the cells in the logarithmic growth phase were used for experiments (*Liu et al., 2020*). HUVECs were treated with 40 µg/ml ox-LDL (Invitrogen LOT2160046, L34357) to form an atherosclerotic model in HUVEC (*Gong et al., 2019*).

## siRNA transfection

HUVECs were cultured in an EC growth medium. Cells passaged two-five times were used for all experiments. The medium was changed every two days, and the cells were used in transfection experiments after seven-ten days (*Simion et al., 2020*). NKILA siRNA and negative control siRNA were lyophilized at 5 nmol each, and 20 μM stock siRNA suspensions were prepared in 250 μl volumes, divided into five aliquots for storage. After treatment with ox-LDL, cells at a density of $1 \times 10^6$/well were incubated in antibiotic-free medium containing siRNA [NKILA siRNA (DH-siRNA; catalog No.: SCC02459) and negative control siRNA (DH-PNC; catalog No.: SCC02408)] and transfection agent (LipoFectMAx catalog No.: FP310, AB1903A2). When the cell density reached 80%, NKILA siRNA (0.005–50 nM) and negative control siRNA (0.005–50 nM) were transfected into cells with lipofectamine (LipoFectMAx catalog No.: FP310, AB1903A2) according to the manufacturer's instructions (*Liu et al., 2020*). In a 2,000 μl/well transfection mixture, volumes of 380–400 μl of serum-free medium, 1,600 μl of complete medium, 10 μl of 5 μM siRNA, and 1.0–10.0 μl of transfection agent were used. After transfection, cells were incubated at 37 °C, 5% CO2 incubator for 24–36 h.

## RNA isolation and quantitative reverse transcription polymerase chain reaction

Total RNA was isolated from HUVECs treated with ox-LDL and siRNA. RNeasy Mini Kit (QIAGEN, catalog no: 74104) was used for RNA isolation. Before starting the experiments, all materials were cleaned and sterilized with nuclease-free water (nzytech MB11101), 70% alcohol solution (obtained from MERCK EMSURE K43900883 242), and RNase away (Sigma, product no: 83931). In this study, BAX, CASP9, CYCS, APAF1, and BCL-2 gene expression in HUVEC treated with ox-LDL and transfected with NKILA siRNA were analyzed using the quantitative reverse transcription polymerase chain reaction (RT-qPCR) method. The experimental design consists of two groups: Group 1: HUVEC treated with ox-LDL and NKILA siRNA (experimental group) and Group 2: HUVEC treated with ox-LDL and negative control siRNA (control group). Each group has five cell culture experiments for quantification of total RNA samples isolated from groups 1 and 2. Total RNA concentration (ng/μl) and purity (A260/A280) of all samples were evaluated by ultraviolet–visible spectroscopy (UV–VIS) (Maestro NANO). The RNA integrity method was not used for samples in this study. There was no inhibition testing in the current study. For the reverse transcription reaction, the RNA concentration of each sample stored at −80 was equalized to 100 ng/μl with nuclease-free water. cDNA synthesis was performed by following the cDNA synthesis kit protocol (RNase Inhibitor High Capacity, catalog no: C03-01-20). Components of the cDNA synthesis reaction include 2 μl of 10X reaction buffer, 1 μl of dNTP mixture, 2 μL of random hexamers, 1 μl of reverse transcriptase, 0.5 μl of RNase inhibitor, 3.5 μL of RNase free water, 10 μL mixture of RNA and RNase free water. A final volume of 20 μl was incubated in the Bio-Rad Thermo Cycler at 25 °C, 37 °C, and 85 °C for 10 min. (min), 120 min., and five min., respectively. We used the SYBR green method to perform RT-qPCR to detect expression levels of genes of interest and control genes in experimental and control groups. SYBR Green is a free-floating

fluorescent dye that binds to double-stranded DNA and increases in fluorescence when bound. In an RT-qPCR reaction, DNA polymerase and primers duplicate the template strand, while SYBR Green binds to the newly formed double-stranded DNA. The RT-qPCR instrument measures the fluorescence, allowing for the quantification of the DNA present in the original sample (*Thermo Fisher Scientific, 2022*). For the RT-qPCR reaction, the RT-qPCR SYBR Green MasterMix kit was used along with the following optimized RT-qPCR primers: APAF1 (gene ID: 317) forward 5′-CCCTTTGTGTCCAGTAGTGGG-3′ and reverse 5′-CTCTGTCTCGCCACATACCC-3′, CASP9 (gene ID: 842) forward 5′-CTTCGTTTCTGCGAACTAACAGG-3′ and reverse 5′-GCACCACTGGGGTAAGGTTT-3′, CYCS (gene ID: 54205) forward 5′-CGCCAATAAGAACAAAGGCATCA-3′ and reverse 5′-TAAGGCAGTGGCCAATTATTACTC-3′, BCL-2 (gene ID: 596) forward 5′-GGATAACGGAGGCTGGGATG-3′ and reverse 5′-TGACTTCACTTGTGGCCCAG-3′, and BAX (gene ID: 581) forward 5′-GATGGACGGGTCCGGGG-3′ and reverse 5′-CGATCCTGGATGAAACCCTGA-3′. In the RT-qPCR step, reactions were repeated three times for both groups. Our study was performed in total with 15 experimental and 15 control group samples. RT-qPCR reaction components were prepared in the following volumes: 6 µl of nuclease-free water, 10 µl of SYBR Green Mastermix (A.B.T.™, cat. no: Q03-02-01), 2 µl of cDNA, 1 µl of forward primer, and 1 µl of reverse primer. Incubation protocol of plates (SEHAGEN) with reaction mixture in Roche LightCycler® 96: Initial incubation, 5 min at 95 °C in 1 cycle; 2-step amplification, 40 cycles: 15 s at 95 °C, 1 min at 60 °C; melting curve analysis, one cycle: 10 s at 95 °C, 1 min at 65 °C, 97 °C for 1 s. Specificity was determined by the melting curve step. The GAPDH gene (gene ID:2597) was used as an internal control (reference gene) for gene expression analysis. The analysis of RT-qPCR expression data was based on ΔΔCt (ΔΔCq) method (Data Analysis Center: https://www.qiagen.com/de/shop/genes-and-pathways/data-analysis-center-overview-page/).

## Statistical analysis

The analysis of RT-qPCR expression data was based on ΔΔCt (ΔΔCq) method (Data Analysis Center: https://www.qiagen.com/de/shop/genes-and-pathways/data-analysis-center-overview-page/). For each gene analyzed in the control and experimental groups, the $P$ value was calculated based on Student's $t$-test. The GAPDH gene (gene ID:2597) was used as an internal control for gene expression analysis. Fold change (FC) is calculated by the formula $2^{-\Delta\Delta Ct}$ and is used to measure the change in the expression level of a gene. Fold regulation (FR) is calculated by logarithmic regulation of FC values and is used to determine how a gene expression is regulated (up-regulation or down-regulation). In the expression analysis, the formulas $\Delta Ct = Ct_{target} - Ct_{reference}$ (normalization), $\Delta\Delta Ct = \Delta Ct_{experimental} - \Delta Ct_{control}$ and $2^{-\Delta\Delta Ct}$ are used, respectively (*Livak & Schmittgen, 2001*).

Pearson correlation test (SigmaPlot 15.0) was used to calculate the correlation between the Ct/Cq (threshold/quantification cycle) values of NKILA expression and the Cq values obtained as a result of the expression of BAX, CASP9, CYCS, APAF1, and BCL-2 genes in ox-LDL and NKILA siRNA-treated HUVEC. $P$ values <0.05 are considered statistically significant.
**Table 1  Fold change (FC), fold regulation (FR) and *P* values of genes expressed in HUVEC transfected with NKILA siRNA after ox-LDL treatment compared to HUVEC transfected with negative control siRNA after ox-LDL treatment.**

| Gene | FC | FR | *P* value |
| --- | --- | --- | --- |
| NKILA | 0.76 | −1.32 | <0.01[*] |
| BAX | 0.75 | −1.34 | 0.32 |
| CASP9 | 0.81 | −1.24 | 0.32 |
| CYCS | 0.74 | −1.35 | 0.33 |
| APAF1 | 0.18 | −5.48 | 0.30 |
| BCL-2 | 0.39 | −2.58 | 0.11 |
| GAPDH | 1.00 | 1.00 | nan |

**Notes.**
   [*]*P* value < 0.05.

# RESULTS

The expression levels of APAF1, CASP9, CYCS, BCL-2, and BAX were analyzed by RT-qPCR in HUVEC treated with ox-LDL and NKILA siRNA. HUVEC was treated with ox-LDL, and negative control siRNA was treated with ox-LDL. Negative control siRNA was used as the control group. When compared with the control group, NKILA gene expression was downregulated 1.32-fold ($P < 0.01^*$) in HUVEC treated with ox-LDL and NKILA siRNA. In comparison to the control group, the expression of BAX (FC = 0.75; $P = 0.32$), CASP9 (FC = 0.81; $P = 0.32$), CYCS (FC = 0.74; $P = 0.33$), APAF1 (FC = 0.18; $P = 0.30$) and BCL-2 (FC = 0.39; $P = 0.11$) in the experimental group was downregulated (Table 1, Fig. 1). There are not any changes in the expression of the reference gene (GAPDH) in the treatment group (HUVEC treated with ox-LDL and NKILA siRNA) compared to the control group (HUVEC treated with ox-LDL and negative control siRNA) in this study. The Cq values of genes in HUVEC transfected with NKILA after ox-LDL treatment were plotted in Fig. 1. According to the results of Pearson correlation analysis between NKILA and apoptotic gene expressions involved in the mitochondrial pathway, there is a positive correlation between NKILA Cq values and Cq values of all genes examined (Table 2). As Cq values of NKILA increased, Cq values of BAX, CASP9, APAF1, and BCL-2 increased at a statistically significant rate ($P < 0.05$) (Table 2).

# DISCUSSION

Atherosclerosis is a major lead of cardiovascular death (*Benjamin et al., 2018*). The main pathology of the disease involves vascular dysfunction, which is induced and exacerbated by inflammation and apoptosis (*Khosravi et al., 2019*; *Wang, Cheng & Chen, 2019*). Recent studies have reported the importance of apoptosis in the pathogenesis of coronary heart disease (*Wu et al., 2018*; *Zhu et al., 2019*). ECs have a crucial role in initiating and progressing atherosclerosis (*Gimbrone Jr & García-Cardeña, 2016*). The endothelium forms a barrier to control the release of immune cells and biomolecules between tissues and circulation. In pathological processes that induce apoptosis, essential biological functions of the endothelium break down (*Werner et al., 2006*). Increased LDL storage and release of circulating leukocytes due to impaired endothelial barrier function initiate the

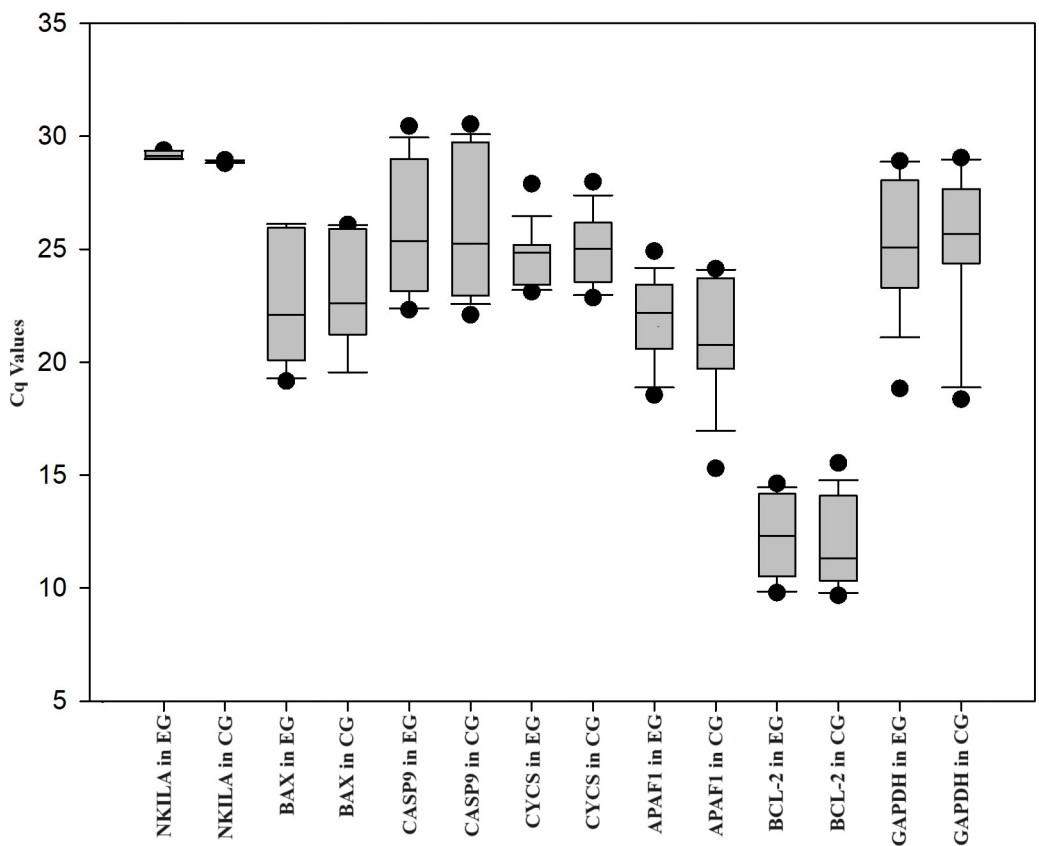

**Figure 1** Box plot of Cq values of genes measured in HUVEC transfected with NKILA siRNA (experimental group, EG) and negative control siRNA (control group, CG) after ox-LDL treatment.

development of atherosclerosis (*Libby et al., 2010*). Increased EC turnover and apoptosis have been reported to occur in atherosclerosis-prone regions of the vascular system and human atherosclerotic plaque endothelium (*Gerrity et al., 1977*; *Tricot et al., 2000*). These findings emphasize the crucial role of EC apoptosis in atherosclerosis development (*Paone et al., 2019*).

Various apoptotic signals, such as hypoxia, cytokines, oxidative stress, and DNA damage, activate the mitochondrial pathway that is related to the lncRNA/messenger RNA (mRNA) axis (*Kalogeris et al., 2012*). Induction of activation of the NF-κB has been releated to resistance to apoptosis. Conversely, NF-κB inhibition causes an increment of apoptosis (*Power, Fanning & Redmond, 2002*). LncRNA NKILA regulates the proinflammatory activities of NF-κB. Induction of NF-κB signaling by inflammation increases expression of NKILA. NF-κB plays an important role in coronary artery disease pathogenesis by altering the endothelial cell functions (*Paone et al., 2019*). Also, lncRNA NKILA may perform as a regulator to inactivate ECs (*Zhu et al., 2019*). In our study, NKILA was silenced with siRNA, and thus, expression of NKILA was downregulated in HUVECs treated with ox-LDL. Expression research has confirmed the downregulation of the NKILA in patients with atherosclerosis compared to the patients in the control group (*Zhu et al., 2019*). The

**Table 2  Correlation between the Cq values for NKILA and the genes studied.**

|  | NKILA<br>r (Pearson correlation coefficient) | *P* value |
|---|---|---|
| BAX | 0.639 | 0.01[*] |
| CASP9 | 0.865 | <0.01[*] |
| CYCS | 0.357 | 0.19 |
| APAF1 | 0.640 | 0.01[*] |
| BCL-2 | 0.622 | 0.01[*] |

**Notes.**

The pair(s) of variables with positive correlation coefficients and *P* values below 0.05 tend to increase together. For the pairs with negative correlation coefficients and *P* values below 0.05, one variable tends to decrease while the other increases. For pairs with *P* values greater than 0.05, there is no significant linear relationship between the two variables.

[*]*P* value < 0.05.

downregulation of lncRNA NKILA is consistent with its proposed anti-inflammatory role (*Ebadi et al., 2022*). It was proposed that lncRNA NKILA promotes cardiomyocyte apoptosis by targeting miR22-3p-TXNIP to inhibit proliferation, migration, and invasion of these cells under high glucose-induced conditions. Today, lncRNA NKILA has been extensively studied for its anticancer effect in several malignancies (*Zhao et al., 2021*). Overexpression of the NKILA was reported to suppress proliferation and induce apoptosis of cervical squamous cell carcinoma cells (*Wang, Zhu & Qiu, 2020*). In cancer studies where NKILA expression was examined in various cell lines, apoptosis increased in studies where NKILA expression increased (*Yu, Tang & Yang, 2018*; *Luo et al., 2020*; *Wang, Zhu & Qiu, 2020*). It has been determined that apoptosis decreases in cells with NKILA degradation (*Bian et al., 2017*; *Gao et al., 2020*).

In the current study, we also found that the expression of pro-apoptotic genes (BAX, CASP9, CYCS, and APAF1) in the HUVEC line treated with ox-LDL and transfected with NKILA siRNA was also decreased.

Intrinsic pathway activation and release of pro-apoptotic proteins from the mitochondria is controlled and regulated by the BCL-2 protein family (*Youle & Strasser, 2008*). Studies have demonstrated that cardiac-specific overexpression of inhibitor of apoptosis (BCL-2) significantly reduces infarction size after ischemia-reperfusion injury (I/R injury) (*Chen et al., 2001*). In our study, BCL-2 gene expression decreased in HUVECs treated with ox-LDL after NKILA silencing. Liu et al. found that increased NKILA expression inhibits NF-κB signaling and attenuates cardiac injury by preventing cell apoptosis and inflammatory responses induced by H/R (hypoxia/reoxygenation) stimulation. This study found that induced NKILA expression reduced the apoptotic cell rate from 29.64% to 22.25%. It also inhibited the expression of caspase-8, caspase-3, caspase-9, and Bax, while positively regulating the anti-apoptotic protein Bcl-2 activated by H/R stimulation (*Liu et al., 2020*). In addition, a group of researchers investigating apoptotic gene expressions to assess whether NKILA can regulate apoptosis of chondrocytes provided evidence that increased expression of NKILA promotes the proliferation of chondrocytes. It was shown that apoptotic genes BAX and CASP3 were significantly downregulated, and anti-apoptotic gene BCL-2 was

significantly upregulated compared to the levels in the control group by upregulation of NKILA (*Xue et al., 2020*). We observed that expression of both pro-apoptotic and anti-apoptotic genes decreased while NKILA expression was downregulated in HUVECs. In another study, NKILA knockdown increased cell viability and suppressed autophagy, cell apoptosis, and inflammation in a lipopolysaccharide-induced sepsis model (*Han et al., 2021*). In diabetic cardiomyopathy patients, ectopic lncRNA NKILA expression enhances cardiomyocyte apoptosis, whereas NKILA knockdown inhibits cardiomyocyte apoptosis (*Li, Li & Su, 2019*). In another study on diabetic cardiomyopathy, it was reported that NKILA was highly expressed in high sugar-induced AC16 cells, and apoptosis protein markers such as BAX, CASP3, and CASP9 were increased, while anti-apoptosis protein BCL-2 was inhibited. NKILA knockdown, on the other hand, had the opposite effect (*Zhao et al., 2021*). In this study, it is thought that lncRNA NKILA may play a role in regulating apoptosis in HUVEC treated with ox-LDL. As a result of NKILA silencing in HUVEC cells, the expression of BAX (FC = 0.75; $P = 0.32$), CASP9 (FC = 0.81; $P = 0.32$), CYCS (FC = 0.74; $P = 0.33$), APAF1 (FC = 0.18; $P = 0.30$) and BCL-2 (FC = 0.39; $P = 0.11$) genes were found to be downregulated. Silencing NKILA may attenuate apoptosis by downregulating the expression of pro-apoptotic genes. However, it may also promote apoptosis due to decreased expression of the anti-apoptotic gene BCL-2. These data suggest that NKILA may be involved in EC apoptosis, but more comprehensive and advanced molecular studies are needed.

The limitations of the study are that protein expression cannot be validated, and apoptosis analysis cannot be performed in cell culture due to limited budget/laboratory facilities.

## CONCLUSION

Atherosclerosis is a disease characterized by EC apoptosis. This study examined how NKILA affects the expression of pro-apoptotic and anti-apoptotic genes in EC cells, suggesting a potential role in regulating cell death. Investigating gene expression involved in apoptosis in HUVECs following ox-LDL treatment and NKILA knockdown will enhance our understanding of the interaction between NKILA and EC apoptosis. This is particularly crucial in the context of atherosclerosis, a disease that contributes significantly to global morbidity and mortality. The novelty of the study is that lncRNA NKILA may play a role in regulating EC apoptosis, which is induced by ox-LDL during the initial stage of atherosclerosis.

### Funding

This work was supported by the Sivas Cumhuriyet University Scientific Research Projects Unit (CUBAP) (grant number: T-2022-964). The funders had no role in study design, data collection and analysis, decision to publish, or preparation of the manuscript.

### Grant Disclosures

The following grant information was disclosed by the authors:
Sivas Cumhuriyet University Scientific Research Projects Unit (CUBAP): T-2022-964.

### Competing Interests

The authors declare there are no competing interests.

### Author Contributions

- Burcu Bayyurt conceived and designed the experiments, performed the experiments, analyzed the data, prepared figures and/or tables, authored or reviewed drafts of the article, and approved the final draft.
- Şeyda Akın performed the experiments, prepared figures and/or tables, authored or reviewed drafts of the article, and approved the final draft.
- Nil Özbilüm Şahin performed the experiments, authored or reviewed drafts of the article, and approved the final draft.
- İzzet Yelkuvan performed the experiments, prepared figures and/or tables, authored or reviewed drafts of the article, and approved the final draft.

### Data Availability

Raw data are available in the Supplemental Files.

### Supplemental Information

Supplemental information for this article can be found online at http://dx.doi.org/10.7717/peerj.17915#supplemental-information.

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
