# Peer review of "Association between NKILA and some apoptotic gene expression in atherosclerosis"

_PeerJ, doi:10.7717/peerj.17915_

## Round 0.1 · original submission · Major Revisions

Your manuscript was considered interesting by the reviewers however they had a number of significant concerns that need to be addressed. First, a clear hypothesis needs to be stated in the abstract and introduction of your manuscript. Additionally, the discussion needs to more clearly explain the effect the changes in gene expression you observed will have on apoptosis, the significance of your findings, as well as future experiments or studies you are planning to perform. One of the reviewers suggested that your results and table 1 need to include the fold change, fold regulation and p-value for GAPDH, and that figure 1 should also include boxplots of the Cq values you measured for each gene in the negative siRNA sample, as well the Cq values for GAPDH. Lastly, more detail needs to be provided as to how fold change and fold regulation were calculated.

Please, submit a detailed rebuttal which shows where and how you have taken all comments and suggestions into consideration. If you do not agree with some of the reviewers’ comments or suggestions, please explain why. Your rebuttal will be critical in making a final decision on your manuscript. Please, note also that your revised version may enter a new round of review by the same or by different reviewers. Therefore, I cannot guarantee that your manuscript will eventually be accepted.

Reviewer 1 ·

Basic reporting

-

Experimental design

-

Validity of the findings

-

Additional comments

In this paper, the authors investigate the association between lncRNA NKILA and apoptosis gene expression in atherosclerosis. They conducted experiments by expressing certain pro- and anti-apoptosis genes in HUVECs treated with ox-LDL, an agent known to induce apoptosis. Following this, they transfected the cells with NKILA-siRNA to silence the lncRNA NKILA. Gene expressions of apoptosis genes were measured by RT-qPCR. The topic is interesting, and the experimental setup is straightforward. I only have some comments to make:


1. In the results section, the authors should also report if there are any changes in the control gene used in this study (GAPDH) in the treatment groups (siRNA) compared to the control group (negative siRNA).
2. The Cq values for GAPDH should be included in Figure 1 to provide a complete picture of the normalization process.
3. In Figure 1, it may be helpful to also show the Cq values of the genes measured in the negative siRNA group to illustrate the expression levels of the genes without NKILA siRNA knockdown and their changes after NKILA siRNA knockdown.
4. GAPDH should also be included in Table 1. The authors should show the fold change (FC), fold regulation (FR), and p-value of GAPDH in the treatment group compared to the control group.
5. The authors should discuss what the changes in apoptosis gene expression mean for apoptosis. Are there any further experiments planned to explore this?
6. The language could be revised more thoroughly for clarity. More detailed information can be included in the abstract section. For example, the acronym "EC" for endothelial cells should be introduced when first mentioned (e.g., EC apoptosis).

Reviewer 2 ·

Basic reporting

Bayyurt and colleagues asked whether NKILA expression is associated with the expression of a number of apoptosis-related genes in HUVEC. The study addresses an important research question, yet it is relatively small and limited in several ways. The English level is not adequate: I respectfully recommend that a peer assists the authors to improve that part. The hypothesis of the study is never clearly expressed in the abstract or in the introduction. What is the degree of novelty of the study? This reviewer is not an expert in NKILA, but the amount of literature on the subject is sizeable. The title is misleading, as HUVEC are not a comprehensive model of the atheroma. The discussion repeats the content of the introduction. What are FR and FC? This reviewer could find any definition of those two variables. It is not clear what the function of Figure 1 is.

Experimental design

I have no comment on the design. If the issues indicated in the section 1. of this review are resolved, the study becomes more intelligible and so does the design.

Validity of the findings

The validity is impossible to assess, as the authors do not explain the methods in detail, particularly the definition of FR and FC

Additional comments

I have no additional comment, thanks.

---

## Round 0.2 · accepted · Accept

Further to the positive re-reviews, I am happy to Accept this manuscript now.

Reviewer 1 ·

Basic reporting

-

Experimental design

-

Validity of the findings

-

Additional comments

All my comments have been addressed in this revision.

Reviewer 2 ·

Basic reporting

I am satisfied with the authors' response. I leave the judgment on the breadth of the study to the Editor, who will speak for the journal's scope.

Experimental design

No observations.

Validity of the findings

No observations.

Additional comments

No additional comments